# Client-only Distributed Markov Chain Monte Carlo Sampling over a Network

**Bo Yuan**                                                          *byuan48@gatech.edu*
*School of Aerospace Engineering*
*Georgia Institute of Technology*

**Jiaojiao Fan**                                                     *jiaojiaf@nvidia.com*
*Nvidia*

**Jiaming Liang**                                              *Jiaming.Liang@rochester.edu*
*Department of Computer Science*
*University of Rochester*

**Yongxin Chen**                                                   *yongchen@gatech.edu*
*School of Aerospace Engineering*
*Georgia Institute of Technology*

**Reviewed on OpenReview:** *https://openreview.net/forum?id=1bZ2rLfKwuG*

## Abstract

We aim to sample from a target $\exp(-\sum_{i=1}^{n} f_i(x|\mathcal{D}_i))$ where each client $f_i$ only has access to local data $\mathcal{D}_i$. We present a fully distributed Markov Chain Monte Carlo (MCMC) sampler that operates through client-to-client communication, eliminating the need for additional centralized servers. Unlike MCMC algorithms that rely on server-client structures, our proposed sampler is entirely distributed, enhancing security and robustness through decentralized communication. In contrast to limited decentralized algorithms arising from Langevin dynamics, our sampler utilizes blocked Gibbs sampling on an augmented distribution. Furthermore, we establish a non-asymptotic analysis of our sampler, employing innovative techniques. This study contributes to one of the initial analyses of the non-asymptotic behavior of a fully distributed sampler arising from Gibbs sampling.

## 1 Introduction

In recent years, inspired by the rapid development in optimization, a lot of sampling algorithms have been proposed and then analyzed to obtain their non-asymptotic behaviors (Dalalyan, 2017b; Dalalyan & Riou-Durand, 2020; Zhang et al., 2023; Fan et al., 2023; Altschuler & Chewi, 2023). In this work, we are interested in sampling from a target distribution of $x \in \mathbb{R}^d$ defined as

$$\pi(x) \propto \exp\left(-\sum_{i=1}^{n} f_i(x|\mathcal{D}_i)\right)$$

in a *distributed* manner. With this formulation, we implicitly have that each potential (negative log density) $f_i$ only possesses access to its local data $\mathcal{D}_i$. Fully distributed systems, e.g., distributed sensor networks (Qi et al., 2001) and edge computation (Mach & Becvar, 2017), provide advantages over centralized ones by enabling computing agents to share limited information, reducing communication costs, and improving security and robustness. These factors make them potentially superior in terms of computational performance and data protection. These raise the need to develop and analyze a distributed sampling algorithm without centralized servers. Our target distribution is the straightforward generalization of the objective function $\sum_{i=1}^{n} f_i(x|\mathcal{D}_i)$ in

distributed optimization, and this problem naturally arises in various applications of distributed optimization, e.g., power systems, sensor networks, and smart manufacturing (Yang et al., 2019).

In the machine learning community, our sampling problem is heavily related to Bayesian learning (Cao et al., 2023), an area carrying on Bayesian inference and uncertain quantification on learned models. Consider a learning setting where the goal is to train models with minimizing $\sum_{i=1}^{n} f_i(x|\mathcal{D}_i)$. As mentioned in Sun et al. (2022), one can view each $f_i$ as a loss function on data collection $\mathcal{D}_i$. Compared with only searching for the minimizer of $\sum_i f_i$, Bayesian inference on the distribution $\pi$ can give more detailed information on the loss landscape. Moreover, Bayesian inference is preferred when only limited data is available; applying prior distributions on model parameters can significantly enhance the performance of models trained on limited data (Sun et al., 2019). In what follows, we omit the dependency on $\mathcal{D}_i$ for brevity.

Many existing distributed Markov Chain Monte Carlo (MCMC) algorithms are deployed with server-client architectures: at each step, the server aggregates information from clients as shown in Figure 1. Distributed sampling via moment-sharing is proposed in Xu et al. (2014). More parallel MCMC algorithms are presented in Wang & Dunson (2013); Neiswanger et al. (2013); Wang et al. (2015); Chowdhury & Jermaine (2018); De Souza et al. (2022). Recently, several distributed MCMC algorithms (Vono et al., 2022b; Plassier et al., 2021; Kotelevskii et al., 2022) are proposed to address practical issues like safety, communication cost, and heterogeneity. These parallel and distributed MCMC algorithms either rely on client-server architectures or lack non-asymptotic convergence rates. To the best of our knowledge, only a handful of sampling literature (Kungurtsev, 2020; Parayil et al., 2020; Gürbüzbalaban et al., 2021; Kolesov & Kungurtsev, 2021; Kungurtsev et al., 2023) proposes fully distributed samplers and establishes the non-asymptotic results. These samplers arise from Langevin Monte Carlo or Hamiltonian Monte Carlo, leveraging the concept of gradient tracking in distributed optimization (Pu & Nedić, 2021). However, unsuitable hyperparameters of these methods may result in failure to sample from the target. As demonstrated in our experiments, samplers based on Langevin Monte Carlo diverge with unreasonably large step sizes, while our proposed sampler ensures convergence with the same step sizes.

Another well-known sampler is the Gibbs sampler, commonly used in asynchronous settings where multiple nodes can update their samples without waiting for each other (Terenin et al., 2020). However, the exploration of the non-asymptotic behaviors of a fully distributed sampler based on *Gibbs sampling* remains uncharted territory. Even though the parallel Gibbs sampler was already proposed in the distributed setting in Gonzalez et al. (2011), we give non-asymptotic results for the first time. Given the intricate nature of Gibbs sampling, which differs from Langevin dynamics, our main challenge is to develop innovative proof ideas to analyze non-asymptotic behaviors.

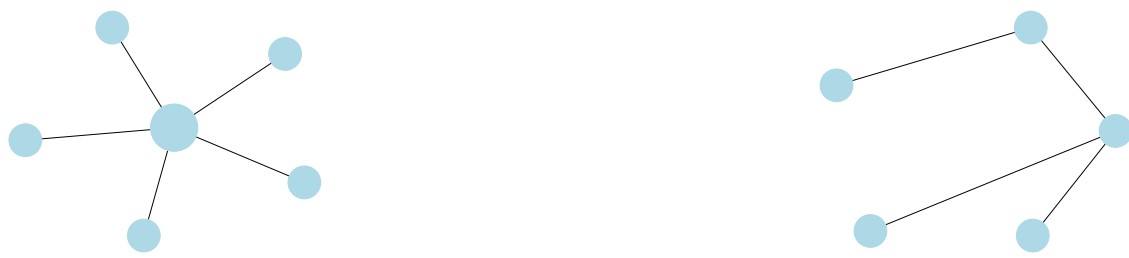

Figure 1: Comparison of server-client architectures with centralized servers (Left) and Trees that enable distributed sampling (Right). In the server-client structure, each client (small node) must communicate with a centralized server (big node).

We fill in the gap with a novel and fully distributed sampling framework. Our work represents one of the *first* endeavors to address the non-asymptotic convergence results of a fully distributed sampling algorithm on a Gibbs sampler. Our experiments further show that, compared with Langevin Monte Carlo baselines, our Gibbs sampler is noticeably more robust to the choice of step size. To utilize a specific Gibbs sampler, we

assume all the clients are distributed on a *bipartite graph*. Note that every connected graph has a spanning tree. For a comparison of the server-client structures and trees, see Figure 1.

To sample from the target $\pi(x) \propto \exp(-\sum_{i=1}^{n} f_i(x))$ where $x \in \mathbb{R}^d$, we instead consider sampling from an augmented distribution of $\{x_i\}_{i=1}^{n} \in \mathbb{R}^{nd}$, $\hat{\pi}(x_1, x_2, \ldots, x_n) \propto \exp(-\sum_{i=1}^{n} f_i(x_i) - \sum_{a \in A} \sum_{b \in B} \sigma_{ab}/2\eta \|x_a - x_b\|^2)$ where $A$ and $B$ stand for two disjoint node groups of a bipartite graph. Here, each $x_i \in \mathbb{R}^d$ stands for one node. We assume $\sigma_{ab} \in (0, 1]$ if there is an edge between $x_a$ and $x_b$, otherwise, $\sigma_{ab} = 0$. Furthermore, $\sum_a \sigma_{ab} \le 1, \sum_b \sigma_{ab} \le 1$ for any $a, b$. The nature of Gibbs sampling, iteratively sampling from conditional distributions, makes it a candidate for distributed sampling. With the specific design of $\hat{\pi}$, it can be observed that if $\eta$ is small enough, the marginals of $\hat{\pi}$ and the distribution of the averaged samples can be arbitrarily close to $\pi$.

A closely related line of research is data-augmentation methods (Vono et al., 2020; 2022a; Plassier et al., 2021), which proposes the joint density $\exp\left(-\sum_{i=1}^{n}(f_i(x_i) + 1/2\eta \|x_i - y\|^2)\right)$ with a new auxiliary variable $y$, and sample from it with a Gibbs scheme. Note that in our augmented distribution, there are no additional nodes involved in the original network, while other data-augmentation algorithms are inherently **centralized**. In contrast, our method is **fully distributed**, offering improved security and robustness compared to centralized samplers.

In terms of the theoretical contributions, Theorems 4.3 and 5.4 deliver new non-asymptotic bias bounds. Our analysis generalizes Corollary 3 in Vono et al. (2020): that corollary is a special case of Theorem 4.3 by setting either $f$ or $g$ to a constant. We extend the result to the two-node setting (Theorem 4.3) and further to bipartite graphs (Theorem 5.4), with new proof ideas.

Our main theoretical contribution is the non-asymptotic bounds of the distance between the average of samples and the target in Theorem 5.4. Combining with improved convergence results of sampling from $\hat{\pi}$ in Yuan et al. (2023), we eventually give the overall iteration complexity in Proposition 5.5. Yuan et al. (2023) only considers the convergence of the Gibbs sampler without quantifying the non-asymptotic error between the original distribution and the marginals. We emphasize that the target distribution in (Yuan et al., 2023) is not the composite target distribution considered in our work. Hence, the nonasympttoic guantee toward to $\exp(-\sum_{i=1}^{n} f_i(x_i))$ does not exist without our contributions in Theorem 4.3 and 5.4. Moreover, we revise the proof of Theorem 10 in (Yuan et al., 2023) and improved the convergence rate as detailed after Theorem 5.3. We also demonstrate our sampler's superior performance under various settings. Please refer to Appendix A.7 for a more detailed discussion on our novelty compared to Yuan et al. (2023).

In summary, our contributions are listed below.

- We propose a new distributed sampling algorithm (Algorithm 2) that does not require a server node and can be deployed on any bipartite graphs. Our sampler is applicable to any target distributions, provided that mild regularity conditions are met (Remark 5.6). Unlike previous works relying on Langevin Monte Carlo or Hamiltonian Monte Carlo, our sampler builds upon Gibbs sampling.

- We analyze its non-asymptotic results (Proposition 5.5) with the bound improved from(Yuan et al., 2023) and the non-trivial upper bound between marginals of $\hat{\pi}$ and $\pi$ (Theorem 5.4) established in this work with novel techniques.

- We conduct both qualitative and quantitative experiments in Section 6 and Appendix A.5, demonstrating the efficiency of our sampler for large-scale distributed systems and the superior performance over the baselines. Specifically, we show that our sampler is markedly more robust than existing Langevin Monte Carlo.

## 2 Related works

**Fully distributed (Decentralized) MCMC with theoretical guarantees** In the following related literature, our fully distributed architecture is typically described as "decentralized". Throughout this paper, we use the term "fully distributed" for our sampler to underscore its distinction from server-dependent distributed

samplers. In Kungurtsev (2020), convergence and consensus guarantee are provided for decentralized Langevin dynamics, with the necessity of a diminishing stepsize for ensuring asymptotic consensus. Gürbüzbalaban et al. (2021) independently investigates decentralized stochastic gradient Langevin dynamics and decentralized stochastic gradient Hamiltonian dynamics under the strong convexity assumption. Parayil et al. (2020) consider unadjusted Langevin algorithm with non-log-concave potentials, replacing convexity assumption with bounded conditions of disagreement on the gradient among the distributed agents. Kolesov & Kungurtsev (2021) considers directed graphs in their analysis. Kungurtsev et al. (2023) shows theoretical guarantees for decentralized Metropolis-adjusted Hamiltonian Monte Carlo, enabling a constant step size. It is worth pointing out that except for Parayil et al. (2020), the convergence rates are quantified with Wasserstein distance, while ours is Total Variation distance, which is due to innovative proof techniques used in this work. Bhar et al. (2023a;b) analyzed decentralized Langevin algorithms under asynchronous and event-triggered conditions, but requires a diminishing step size, while we focus on synchronous settings and utilize a constant step size. Tzikas et al. (2024) is, to the best of our knowledge, the most recent work on decentralized MCMC. Its theoretical guarantees, however, are weaker than ours. Assumption 5 is uncommon, and Theorem 2 predicts a counter-intuitive outcome: the sampler converges more slowly as the network becomes denser. In contrast, our analysis relies on standard assumptions, and Proposition 5.5 shows that convergence speeds up with increased connectivity. Our experiments further show the superior performance of our sampler.

See Appendix A.1 for more related works.

## 3 Preliminaries

### 3.1 Problem setting

Throughout this paper, we aim to perform Bayesian inference on distributed models by sampling from a target distribution

$$\pi(x) \propto \exp\left(-\sum_{i=1}^{n} f_i(x)\right) \tag{1}$$

where $x \in \mathbb{R}^d$. In these models, data is stored locally at each client. To uphold privacy, only estimated parameters are permitted to traverse across the clients.

Without loss of generality, we assume that all the clients are located on an undirected tree structure. This assumption can be satisfied as long as all the clients are on an undirected connected graph. To obtain the non-asymptotic convergence bounds, we utilize the standard assumptions on each potential: we assume each $f_i$ is $\alpha_i$-strongly-convex and $\beta_i$-smooth. This condition implies that for any $x$ and $y$,

$$\nabla f_i(y)^T (x - y) + {\alpha_i}/{2}\|x - y\|^2 \leq f_i(x) - f_i(y),$$
$$\nabla f_i(y)^T (x - y) + {\beta_i}/{2}\|x - y\|^2 \geq f_i(x) - f_i(y).$$

### 3.2 Notation

For a joint distribution $\pi^{XY}$ defined on $(X, Y)$, the $X$-marginal distribution is represented by $\pi^X$, and the $X$-conditional distribution given $y$ is $\pi^{X|Y=y}$. The iteration complexity is the required number of iterations of Gibbs samplers to achieve a given accuracy.

### 3.3 Sampling background

Here we list several useful tools and lemmas we utilize. The Total Variation (TV) distance and the Kullback-Leibler (KL) divergence are widely used to quantify the distance between two distributions. For two measures $\mu$ and $\nu$ with density function representations, we have $\text{TV}(\mu, \nu) = \frac{1}{2}\|\mu - \nu\|_1$. The KL divergence of $\mu$ and $\nu$ is defined as $\text{KL}(\mu\|\nu) = \int \log \frac{\mu}{\nu}\mu$. Pinsker's inequality states that $\text{TV}(\mu, \nu) \leq \sqrt{\frac{1}{2}\text{KL}(\mu\|\nu)}$. We also quantify the convergence with $W_2$ distance in experiments. It is defined as $W_2(\mu, \nu) = \inf_\pi (\mathbf{E}_\pi \|x - y\|^2)^{1/2}$ where $\pi$ is the coupling of $\mu$ and $\nu$.

Recall that a probability measure $\nu$ satisfies Logarithmic Sobolev Inequality (LSI) with a constant $\alpha > 0$ (in short, LSI($\alpha$)), if for any distribution $\mu$ such than $\mu \ll \nu$, the KL divergence $\mathrm{KL}(\mu||\nu)$ and Fisher information $I(\mu||\nu) = \int \|\nabla \log \frac{\mu}{\nu}\|^2 \mu$ satisfy $\mathrm{KL}(\mu||\nu) \leq \frac{1}{2\alpha} I(\mu||\nu)$. It is well-known that a strongly log-concave distribution $\nu$ with a constant $\alpha$, equivalently its potential is $\alpha$-strongly-convex, satisfies LSI with the same constant. Moreover, we utilize the following lemma in our proof.

**Lemma 3.1** (Theorem 1 in Durmus & Moulines (2019)). *For a strongly log-concave distribution $\pi$ with $\alpha$ defined on $\mathbb{R}^d$, one has $\mathbf{E}_{x \sim \pi} \|x - m\|^2 \leq \frac{d}{\alpha}$ where $m$ is the minimizer of the potential.*

# 4   Analysis of Composite Sampling

In this section, we aim to provide the analysis for composite sampling, i.e., sampling from a special case of equation 1,

$$\pi(x) \propto \exp(-f(x) - g(x)). \tag{2}$$

We emphasize that in this distributed setting, each client symmetrically generates samples from conditional distributions, without the need for an additional server to aggregate parameters from the two clients. This two-node composite sampling is a special case of equation 1, and our main analysis of distributed sampling is essentially a generalization of ideas in this section. To proceed, we have the following assumption.

**Assumption 4.1.** Let $f(x)$ be $\alpha_f$-strongly-convex and $\beta_f$-smooth. Similarly, let $g(x)$ be an $\alpha_g$-strongly-convex and $\beta_g$-smooth function. It is also necessary to assume $f$ and $g$ share the same minimizer $m$.

Inspired by the success of proximal samplers, this sampling task can be performed by sampling from an augmented distribution

$$\pi^{XY}(x, y) \propto \exp\left(-f(x) - g(y) - \frac{1}{2\eta}\|x - y\|^2\right)$$

and then control the TV distance between $\pi$ and $\pi^X$ with a sufficiently small $\eta$. The procedure is summarized in Algorithm 1. We emphasize that Steps 2–4 follow the standard Gibbs sampling procedure, as also adapted in Yuan et al. (2023). Compared to Yuan et al. (2023), one of our main contributions is to apply the standard Gibbs sampling procedure on a different target distribution and obtain the non-asymptomatic convergence rate to the new target.

The strong convexity and smoothness assumptions are standard and have been used in a series of sampling works (Dalalyan, 2017a; Dalalyan & Karagulyan, 2019; Fan et al., 2023; Altschuler & Chewi, 2023). Note that the other assumption that $f$ and $g$ have the same minimizer is required to establish the non-asymptotic bounds of $\mathrm{TV}(\pi, \pi^X)$. The same assumption has been discussed in Lee et al. (2021) and Yuan et al. (2023). In Lee et al. (2021), this assumption is necessary to analyze the non-asymptotic results of samplers for the same target equation 2. As mentioned in Lee et al. (2021), for two functions $\hat{f}$ and $\hat{g}$ that have different minimizers, one can easily find $m$ such that $\nabla \hat{f}(m) + \nabla \hat{g}(m) = 0$ since both $f$ and $g$ are strongly convex. Then define $f(x) := \hat{f}(x) - \langle \nabla f(m), x \rangle$ and $g(x) := \hat{g}(x) - \langle \nabla g(m), x \rangle$. It is easy to verify that sampling from $\exp(-f(x) - g(x))$ is equivalent to the original target, and $f$ and $g$ share the same minimizer $m$. Furthermore, the necessity of this assumption has been illustrated in Example 3 in Yuan et al. (2023). It is worth emphasizing that Example 3 shows that, without the same minimizer assumption, the non-asymptotic bound also depends on other parameters: when $f$ and $g$ are quadratic, it also relies on the distance between the centers of the two functions.

The convergence result of the Gibbs sampler in Algorithm 1 has been established in Theorem 5 in Yuan et al. (2023). For the sake of completeness, we restate it here.

**Theorem 4.2** (Theorem 5 in Yuan et al. (2023)). *Assume Step 3 and Step 4 are exact without any bias, i.e., $y^k$ and $x^{k+1}$ follow the conditional distributions exactly. Denote the distribution of the $k$-th samples $(x^k, y^k)$ by $\nu_k^{XY}$. For the target distribution $\pi^{XY}$, under Assumption 4.1, we have*

$$\mathrm{KL}(\nu_{k+1}^X || \pi^X) \leq \mathrm{KL}(\nu_k^X || \pi^X) \frac{1}{(1 + \eta \alpha_f)^2 (1 + \eta \alpha_g)^2},$$

*where $\nu_k^X$ is the X-marginal of $\nu_k^{XY}$.*

---

**Algorithm 1** A Sampler for Composite Potentials

---

**Input:** $\pi : \exp(-f(x) - g(x))$: target distribution
**Output:** $x^K$: Samplers within $\epsilon$ TV distance of $\pi$
Step 1: Construct a distribution $\pi^{XY} : \exp(-f(x) - g(y) - \frac{1}{2\eta}\|x - y\|^2)$
\# The following is the Gibbs sampler for $\pi^{XY}$
Step 2: Generate an initial sample $x^0$ from $\nu_0^X$
**for** $k \leftarrow 0, \cdots, K - 1$ **do**
  Step 3: Sample $y^k \sim \pi^{Y|X=x^k}(y) \propto \exp(-g(y) - \frac{1}{2\eta}\|x^k - y\|^2)$
  Step 4: Sample $x^{k+1} \sim \pi^{X|Y=y^k}(x) \propto \exp(-f(x) - \frac{1}{2\eta}\|x - y^k\|^2)$
**end for**

---

To obtain the overall iteration complexity of Algorithm 1, it remains to select a suitable $\eta$ based on $\mathrm{TV}(\pi, \pi^X)$, which can be seen from Theorem 4.3. Even though similar intuition has been discussed in Yuan et al. (2023), our work rigorously proves the non-asymptotic bound of $\mathrm{TV}(\pi, \pi^X)$ for the first time.

**Theorem 4.3.** *Under Assumption 4.1, we have*

$$\mathrm{TV}(\pi, \pi^X) \le \sqrt{\frac{\beta_g^2 d}{2(\alpha_f + \frac{\alpha_g}{\eta\alpha_g+1})}\left(\frac{\eta^2\beta_g^2}{\alpha_f + \alpha_g} + \frac{\eta}{\eta\alpha_g + 1}\right)} \tag{3}$$

*Proof.* As both $f$ and $g$ are strongly convex, by Theorem 3.8. in Saumard & Wellner (2014), the $X$-marginal distribution $\pi^X$ is strongly-log-concave with a parameter $\alpha_f + \frac{\alpha_g}{\eta\alpha_g+1}$. Hence, $\pi^X$ satisfies $\mathrm{LSI}(\alpha_f + \frac{\alpha_g}{\eta\alpha_g+1})$. This implies that

$$\mathrm{KL}(\pi\|\pi^X) \le \frac{1}{2(\alpha_f + \frac{\alpha_g}{\eta\alpha_g+1})}I(\pi\|\pi^X) \tag{4}$$

By definition, the relative Fisher information $I(\pi\|\pi^X)$ satisfies

$$\begin{aligned}
I(\pi\|\pi^X) &= \mathop{\mathbf{E}}_{x\sim\pi}\left\|\nabla\log\frac{\pi(x)}{\pi^X(x)}\right\|^2 \\
&= \mathop{\mathbf{E}}_{x\sim\pi}\|\nabla g(x) + \nabla\log G_\eta(x)\|^2,
\end{aligned} \tag{5}$$

where $G_\eta(x) = \int \exp(-g(y) - \frac{1}{2\eta}\|x - y\|^2)dy$. To proceed, notice that

$$\nabla_x \log G_\eta(x) = \mathbf{E}_{y\sim\exp(-g(y)-\frac{1}{2\eta}\|x-y\|^2)}(-\nabla g(y)) \tag{6}$$

where we use the identity $\int\left(\nabla_y \exp\left(-g(y) - \frac{1}{2\eta}\|x - y\|^2\right)\right)dy = 0$. Combining equation 5 and equation 6 yields that

$$\begin{aligned}
&I(\pi\|\pi^X) \\
&= \mathbf{E}_{x\sim\pi}\left\|\mathbf{E}_{y\sim\exp(-g(y)-\frac{1}{2\eta}\|x-y\|^2)}(\nabla g(x) - \nabla g(y))\right\|^2 \\
&\le \mathbf{E}_{x\sim\pi}\mathbf{E}_{y\sim\exp(-g(y)-\frac{1}{2\eta}\|x-y\|^2)}\|\nabla g(x) - \nabla g(y)\|^2 \\
&\le \beta_g^2\mathbf{E}_{x\sim\pi}\mathbf{E}_{y\sim\exp(-g(y)-\frac{1}{2\eta}\|x-y\|^2)}\|x - y\|^2
\end{aligned} \tag{7}$$

where we adopt Jensen's inequality and the smoothness of $g$. Then we estimate the upper bounds of two expectations separately. Denote the minimizer of $g(y) + \frac{1}{2\eta}\|x - y\|^2$ as $y^*$. For the inner expectation over $y$ in equation 7, we have

$$\begin{aligned}
&\mathbf{E}_{y\sim\exp(-g(y)-\frac{1}{2\eta}\|x-y\|^2)}\|x - y\|^2 \\
&\le 2\|x - y^*\|^2 + 2\mathbf{E}_{y\sim\exp(-g(y)-\frac{1}{2\eta}\|x-y\|^2)}\|y^* - y\|^2 \\
&\le 2\|x - y^*\|^2 + \frac{2d}{\alpha_g + 1/\eta}.
\end{aligned} \tag{8}$$

Here, the last line is due to Lemma 3.1. Since $\nabla g(y^*) + \frac{1}{\eta}(y^* - x) = 0$, for the outer expectation over $x$ in equation 7,

$$\mathbf{E}_{x \sim \pi} \|x - y^*\|^2 = \eta^2 \mathbf{E}_{x \sim \pi} \|\nabla g(y^*)\|^2 \tag{9}$$
$$\leq \eta^2 \beta_g^2 \mathbf{E}_{x \sim \pi} \|y^* - m\|^2$$
$$\leq \eta^2 \beta_g^2 \mathbf{E}_{x \sim \pi} \|x - m\|^2 \leq \eta^2 \beta_g^2 \frac{d}{\alpha_f + \alpha_g} \tag{10}$$

The second inequality is due to the smoothness of $g$ and the assumption that $\nabla g(m) = 0$, and the third inequality is from the contraction property of the proximal method in optimization. For the last line, we use Lemma 3.1 and the assumption that $f$ and $g$ have the same minimizer. With equation 4, equation 7, equation 8 and equation 9, one has

$$\mathrm{KL}(\pi \| \pi^X) \leq \frac{\beta_g^2 d}{\alpha_f + \frac{\alpha_g}{\eta \alpha_g + 1}} \left( \frac{\eta^2 \beta_g^2}{\alpha_f + \alpha_g} + \frac{\eta}{\eta \alpha_g + 1} \right) \tag{11}$$

By Pinsker's inequality, we obtain equation 3. $\qquad \square$

Combining Theorem 4.2 and Theorem 4.3, we are ready to show the overall iteration complexity in Proposition 4.4.

**Proposition 4.4.** *For simplicity, we assume $\eta$ is small enough that the second order term $\eta^2 \beta_g^2/(\alpha_f + \alpha_g)$ in Theorem 4.3 is upped bounded by the first order term $\frac{\eta}{\eta \alpha_g + 1}$, and $\eta \alpha_g < 1$. Under Assumption 4.1, to achieve $\epsilon$ TV distance to $\pi$, with Algorithm 1 where $\eta = \boldsymbol{O}\left( \frac{(\alpha_f + \alpha_g)\epsilon^2}{(\beta_f + \beta_g)^2 d} \right)$, one needs iterations*

$$K = \boldsymbol{O}\left( \frac{(\beta_f + \beta_g)^2 d}{(\alpha_f + \alpha_g)^2 \epsilon^2} \log\left( \frac{\mathrm{KL}(\nu_0^X \| \pi^X)}{\epsilon^2} \right) \right). \tag{12}$$

*Proof.* Notice that the overall biased term $\epsilon$ comes from two components: the Gibbs sampler for $\pi^{XY}$ and the distance between $\pi$ and $\pi^X$.

By Theorem 4.2, the number of iterations to achieve $\mathrm{TV}(\nu_K^X, \pi^X) \leq \epsilon/2$ is

$$K = \boldsymbol{O}\left( \log\left( \frac{\mathrm{KL}(\nu_0^X \| \pi^X)}{\epsilon^2} \right) \frac{1}{\eta(\alpha_f + \alpha_g)} \right). \tag{13}$$

To ensure $\mathrm{TV}(\pi, \pi^X) \leq \epsilon/2$, combining the assumption that $\frac{\eta^2 \beta_g^2}{\alpha_f + \alpha_g}$ is is bounded by the first order term and Theorem 4.3, one needs

$$\eta = \boldsymbol{O}\left( \frac{(\alpha_f + \alpha_g)\epsilon^2}{\beta_g^2 d} \right). \tag{14}$$

Therefore, to have $\mathrm{TV}(\tilde{\nu}_T^X, \pi) \leq \epsilon$, with equation 13 and equation 14, $\eta$ and $K$ can be $\eta = \boldsymbol{O}\left( \frac{(\alpha_f + \alpha_g)\epsilon^2}{(\beta_f + \beta_g)^2 d} \right)$, and $K = \boldsymbol{O}\left( \frac{(\beta_f + \beta_g)^2 d}{(\alpha_f + \alpha_g)^2 \epsilon^2} \log\left( \frac{\mathrm{KL}(\nu_0^X \| \pi^X)}{\epsilon^2} \right) \right)$. $\qquad \square$

*Remark* 4.5 (Weaker Assumptions). The result of Proposition 4.4 can be generalized to the case that $g$ is still strongly convex and smooth, but $f$ is convex and smooth. Then, with the same procedure, one can show the iteration complexity is $K = \boldsymbol{O}\left( \frac{(\beta_f + \beta_g)^2 d}{\alpha_g^2 \epsilon^2} \log\left( \frac{\mathrm{KL}(\nu_0^X \| \pi^X)}{\epsilon^2} \right) \right)$.

## 5 Analysis of Distributed Sampling over bipartite graphs

In this section, we present our main result: a distributed sampler and its non-asymptotic analysis. Recall that we are interested in sampling from

$$\pi(x) \propto \exp\left( -\sum_{i=1}^{n} f_i(x) \right) \tag{15}$$

with the following Assumption 5.1.

**Assumption 5.1.** Without loss of generality, assume that each $f_i(x)$ is $\alpha$-strongly convex and $\beta$-smooth. We also assume that each $f_i$ shares the same minimizer $m$. As explained in Section 4, this assumption is necessary to have non-asymptotic results, and our sampler still converges asymptotically even without this assumption, as discussed in Remark 5.6.

Note the assumption on the same minimizers is computationally feasible to satisfy under the strong convex and smoothness assumption. As discussed above Algorithm 1, potentials with different minimizers can be transformed to satisfy the same minimizer assumption without altering the target. This transformation mainly involves computing minimizer of $\sum_i f_i$. The computation cost of this step is negligible comparing with the sampling iterations when the potentials are strongly convex and smooth. For example, efficient methods such as fast incremental gradient methods and randomized algorithms (Shalev-Shwartz & Zhang, 2014; Nitanda, 2014; Lin et al., 2015; Lan & Zhou, 2018) have been developed to solve strongly convex and smooth finite-sum problems in a distributed manner. These methods exhibit complexity that depends on the square root of the condition number, independent of the dimension.

It is worth noting that the bipartite graph assumption can be fulfilled on any connected graph. For example, every connected graph has a spanning tree (Serre, 2002), and a tree is always a bipartite graph (Bondy et al., 1976). In this setting, the sampler and its iteration complexity are established with steps similar to Section 4. To begin with, denote

$$\hat{\pi}^{AB}(x_1, x_2, \ldots, x_n) \propto \exp\left( -\sum_{i=1}^{n} f_i(x_i) - \sum_{a \in A} \sum_{b \in B} \sigma_{ab}/2\eta \|x_a - x_b\|^2 \right). \tag{16}$$

With a slight abuse of notation, we will interchangeably use $\hat{\pi}^{AB}$ and $\hat{\pi}$. We have the following assumption for the weights $\sigma_{ab}$.

**Assumption 5.2.** Let $\sigma_{ab} > 0$ stand for there is an edge between nodes $a$ and $b$. Otherwise, $\sigma_{ab} = 0$. Moreover, we assume $\sum_a \sigma_{ab} \leq 1$ and $\sum_b \sigma_{ab} \leq 1$. Furthermore, we assume the graph is a fully connected bipartite graph. Define the $n$ by $n$ doubly stochastic matrix $\Sigma$ whose element at the $s$-th row and $t$-th column is $\sigma_{st}$. This matrix satisfies $\sigma_{ts} = \sigma_{st} = \sigma_{ab} \geq 0$ if $s = a$ and $t = b$, and $\sigma_{ss} = 1 - \sum_a \sigma_{sa} \geq 0$. Let $(\lambda_1, \ldots, \lambda_n) \in \mathbb{R}^n$ be the eigenvalues of $\Sigma$ organizing in the descending order, i.e., $\lambda_i \geq \lambda_{i+1}$,

A bipartite graph divides all nodes into two independent sets: $A$ and $B$. One can view $f_i$ as a local distribution on each node, and the quadratic terms $\|x_a - x_b\|^2$ correspond to Gaussian distributions on edges. With the same idea in Section 4, one can sample from $\hat{\pi}$ with Gibbs sampling and then reduce the distance between the target and the distribution of $\sum_{i=1}^{n} x_i/n$ with a small $\eta$. To sample from $\hat{\pi}$, it is natural to use a block Gibbs sampler where, at each iteration, each set of parameters is updated simultaneously as shown in Algorithm 2. Namely, in Step 3-4, one can generate the $k$-th samples, $\{x_b^k\}_{b \in B}$ and $\{x_a^k\}_{a \in A}$, *in parallel*. We emphasize that, thanks to the bipartite graph structure, our sampler is fully distributed without the need for server nodes. In practice, each node only receives information from its neighbors. As explained in 4, While Yuan et al. (2023) also employs Gibbs updates, our analysis concerns a different target distribution and, crucially, establishes a non-asymptotic rate for this new target.

To obtain the overall iteration complexity of the sampler in Algorithm 2, we start with the non-asymptotic bounds associated with the block Gibbs sampler. Theorem 10 in Yuan et al. (2023) establishes the convergence rate of the block Gibbs sampler over bipartite graphs. We improve the original convergence rates after reviewing its proof. The improvement comes from both the definition of $\hat{\pi}^{AB}$ and a tighter upper bound of $c_t$ in Equation (12) in Yuan et al. (2023). Notice that in Yuan et al. (2023), $\sigma_{ab} \in [0, 1]$ without the additional constrain that $\sum_a \sigma_{ab} \leq 1, \sum_b \sigma_{ab} \leq 1$ we add in this work. Hence, the lower bound of $\eta_j$ used in the proof of Theorem 10 in Yuan et al. (2023), which is defined as $\eta/\sum_a \sigma_{ab}$ with our notation, can be improved to $\eta$. This leads to Lemma A.1. Directly applying our improved results yields the following convergence rate. The proof of Theorem 5.3 is in Appendix A.2.

**Theorem 5.3** (Improved from Theorem 10 in Yuan et al. (2023)). *Assume Steps 3 and 4 in Algorithm 2 are exact without biased terms. Denote the distribution of the samples generated in the $k$-th iteration by $\nu_k^{AB}$.*

---

**Algorithm 2** Distributed Sampling over a bipartite graph

---

    **Input:** $\pi$ as in equation 1: the target distribution
    **Output:** $\{x_a^K\}_{a \in A}$ and $\{x_b^K\}_{b \in B}$
    Step 1: Construct a distribution $\hat{\pi}^{AB}$ as in equation 16
    # The following is the block Gibbs sampler for $\hat{\pi}^{AB}$
    Step 2: Generate the initial samples $\{x_a^A\}_{a \in A}$ from $\nu_0^A$.
    **for** $k \leftarrow 0, \cdots, K-1$ **do**
        Step 3: Sample $\{x_b^k\}_{b \in B} \sim \hat{\pi}^{\{x_b\}|\{x_a^k\}_{a \in A}}$
        which is proportional to $\exp(-\sum_{b \in B} f_b(x_b) - \sum_{a \in A} \sum_{b \in B} \frac{\sigma_{ab}}{2\eta} \|x_a^k - x_b\|^2))$
        Step 4: Sample $\{x_a^{k+1}\}_{a \in A} \sim \hat{\pi}^{\{x_a\}|\{x_b^k\}_{b \in B}}$
        which is proportional to $\exp(-\sum_{a \in A} f_a(x_a) - \sum_{a \in A} \sum_{b \in B} \frac{\sigma_{ab}}{2\eta} \|x_a - x_b^k\|^2))$
    **end for**

---

*For the target distribution $\hat{\pi}^{AB}$ equation 16, under Assumption 5.1, we have*

$$\mathrm{KL}(\nu_k^{AB} \| \hat{\pi}^{AB}) \le \exp(-kC)\mathrm{KL}(\nu_0^A \| \hat{\pi}^A) \tag{17}$$

*where $C$ is*

$$
\begin{aligned}
C = \eta \int_0^1 \Bigg( & \left[ \frac{\eta t(1-t)}{\min_b \sum_a \sigma_{ab}} + \frac{(1-t)^2 |B|}{\alpha} + \frac{t^2}{\alpha} \right]^{-1} \\
& + \left[ \frac{\eta t(1-t)}{\min_a \sum_b \sigma_{ab}} + \frac{(1-t)^2 |A|}{\alpha} + \frac{t^2}{\alpha} \right]^{-1} \Bigg) \, \mathrm{d}t.
\end{aligned}
\tag{18}
$$

*Here $|A|$ and $|B|$ are the number of nodes in the two disjoint groups, respectively.*

As a comparison, in Yuan et al. (2023), the constant $C$ is

$$
\begin{aligned}
\int_0^1 \Bigg( & \frac{\eta}{|A|} \left[ \frac{\eta t(1-t)}{\min_b \sum_a \sigma_{ab}} + \frac{(1-t)^2 |A||B|}{\alpha} + \frac{t^2}{\alpha} \right]^{-1} \\
& + \frac{\eta}{|B|} \left[ \frac{\eta t(1-t)}{\min_a \sum_b \sigma_{ab}} + \frac{(1-t)^2 |A||B|}{\alpha} + \frac{t^2}{\alpha} \right]^{-1} \Bigg) \, \mathrm{d}t.
\end{aligned}
$$

This implies that, for the Gibbs sampler analyzed in Theorem 5.3, our new bound can enjoy an $\mathcal{O}(n^2)$ improvement, where $n$ is the number of nodes.

Now we show the proof on bounding $\mathrm{TV}(\bar{\pi}, \pi)$ where $\bar{\pi}$ is the distribution of $\sum_i x_i / n$ with joint distribution being $\hat{\pi}$. The following Theorem 5.4 holds for any graph, so for the sake of presentation, we rewrite

$$\hat{\pi}(x_1, x_2, \ldots, x_n) \propto \exp\left( -\sum_{i=1}^n f_i(x_i) - \sum_{s=1}^n \sum_{t=1}^n \sigma_{st}/2\eta \|x_s - x_t\|^2 \right). \tag{19}$$

**Theorem 5.4.** *Under Assumption 5.1 on the potentials and Assumption 5.2 on the weights, we have*

$$\mathrm{TV}(\bar{\pi}, \pi) = \boldsymbol{O}\left( \sqrt{\frac{n\beta^2 d}{\alpha} \left( \frac{(n-1)}{\alpha + \frac{n}{\eta}(1-\lambda_2)} + \frac{\eta^2 \beta^2}{(1-\lambda_2)^2 \alpha} \right)} \right). \tag{20}$$

*where $\bar{\pi}$ is the distribution of $\sum_i x_i / n$ given the joint distribution $\hat{\pi}$ in equation 19, and $\lambda_2$ is the second largest eigenvalue of the doubly stochastic matrix $\Sigma$ defined in Assumption 5.2*

*Proof.* First, notice that $\sum_{s=1}^n \sum_{t=1}^n \sigma_{st}/2\eta \|x_s - x_t\|^2 = \frac{1}{2\eta} X^T (\mathbf{I}_{nd} - \Sigma \otimes \mathbf{I}_d) X$ where $X^T = (x_1^T, \ldots, x_n^T)$.

It is well-known that the eigenvalues of $\Sigma$, denoted as $(\lambda_1, \ldots, \lambda_n)$ in Assumption 5.2, are from $-1$ to $1$, and $(1, \ldots, 1)$ is one eigenvector corresponding to eigenvalue $1$. Define the eigendecomposition of $\mathbf{I}_{nd} - \Sigma \otimes \mathbf{I}_d$ as $U \Lambda U^T$. Note that the eigenvalues of $\mathbf{I}_{nd} - \Sigma \otimes \mathbf{I}_d$ are $1 - \lambda_i$ for each $\lambda_i$ in the set of eigenvalues of $\Sigma$. Since we assume the graph is fully connected, there is only one solution to $1/2\eta X^T(\mathbf{I}_{nd} - \Sigma \otimes \mathbf{I}_d)X = 0$, which is $x_1 = x_2 \ldots = x_n$. This implies that $1 - \lambda_1 = 0$ and $1 - \lambda_i > 0$ for $i = 2, \ldots, n$.

With the eigendecomposition, the potential of $\hat{\pi}$ is $F(\sqrt{n}U\hat{X}) + \frac{n}{2\eta}\hat{X}^T \Lambda \hat{X}$. Here, $F(X) := \sum_{i=1}^n f_i(x_i)$ and $\hat{X} := \frac{1}{\sqrt{n}}U^T X$. As the first $d$ columns of $\sqrt{n}U$ are $(1, \ldots, 1)^T \otimes \mathbf{I}_d$, we have $\hat{x}_1 = \frac{1}{n}\sum_{i=1}^n x_i$. Then, the distribution over the average $\frac{1}{n}\sum_{i=1}^n x_i$ is

$$\hat{\pi}^{\hat{X}_1}(\hat{x}_1) \propto \int \exp(-F(\sqrt{n}U\hat{X}) - n/2\eta \hat{X}^T \Lambda \hat{X}) \prod_{i=2}^n \mathrm{d}\hat{x}_i. \tag{21}$$

Intuitively, when $\eta$ is small enough, $\hat{X}^T \Lambda \hat{X}$ should be approximately zero. Since $1 - \lambda_i = 0$ if and only if $i = 1$, $\hat{\pi}^{\hat{X}_1}$ should converge to $\exp(-F(\sqrt{n}U\hat{X}))|_{\hat{X}^T = (\hat{x}_1^T, \mathbf{0}, \ldots, \mathbf{0})}$ as $\eta$ goes to $0$. With this intuition, we have the following rigorous proof.

When $\hat{X}^T = (\hat{x}_1^T, \mathbf{0}, \ldots, \mathbf{0})$, one has

$$F(\sqrt{n}U\hat{X}) = \sum_{i=1}^n f_i(\hat{x}_1), \tag{22}$$

which is exactly the potential of the target $\pi$. Hence, it is sufficient to bound the TV distance between distributions in equation 21 and equation 22. This is achieved following similar proof ideas used in the proof of Theorem 4.3.

Under Assumption 5.1, by Theorem 3.8. in Saumard & Wellner (2014), the $\hat{X}_1$-marginal distribution of $\hat{\pi}(\hat{X})$ is strongly-log-concave with a parameter $n\alpha$. Hence, $\pi^{\hat{X}_1}$ satisfies LSI($n\alpha$). This implies that

$$\mathrm{KL}(\pi||\hat{\pi}^{\hat{X}_1}) \leq \frac{1}{2n\alpha} I(\pi||\hat{\pi}^{\hat{X}_1}). \tag{23}$$

By definition and Jensen's inequality,

$$\begin{aligned}
I(\pi||\pi^{\hat{X}_1}) &= \mathbf{E}_\pi \left\| \nabla_{\hat{x}_1} \log \frac{\pi}{\hat{\pi}^{\hat{X}_1}} \right\|^2 = \mathbf{E}_\pi \left\| \nabla_{\hat{x}_1} \log \frac{\hat{\pi}^{\hat{X}_1}}{\pi} \right\|^2 \\
&= \mathbf{E}_\pi \| \nabla_{\hat{x}_1} F(\sqrt{n}U\hat{X})|_{\hat{X}=(\hat{x}_1, \mathbf{0}, \ldots, \mathbf{0})} \\
&\quad - \mathbf{E}_{\hat{\pi}^{\hat{X}_1=\hat{x}_1}} \nabla_{\hat{x}_1} F(\sqrt{n}U\hat{X})|_{\hat{X}=(\hat{x}_1, \hat{x}_2, \ldots, \hat{x}_n)} \|^2 \\
&\leq \mathbf{E}_\pi \mathbf{E}_{\hat{\pi}^{\hat{X}_1=\hat{x}_1}} \| \nabla_{\hat{x}_1} F(\sqrt{n}U\hat{X})|_{\hat{X}=(\hat{x}_1, \mathbf{0}, \ldots, \mathbf{0})} \\
&\quad - \nabla_{\hat{x}_1} F(\sqrt{n}U\hat{X})|_{\hat{X}=(\hat{x}_1, \hat{x}_2, \ldots, \hat{x}_n)} \|^2.
\end{aligned}$$

Here $\hat{\pi}^{\hat{X}_1=\hat{x}_1}$ is the conditional distribution of $\hat{\pi}$ given $\hat{x}_1$, i.e.,

$$\exp\left( -F(\sqrt{n}U\hat{X})|_{\hat{X}=(\hat{x}_1, \hat{x}_2, \ldots, \hat{x}_n)} - \frac{n}{2\eta}\sum_{i=2}^n (1 - \lambda_i)\|\hat{x}_i\|^2 \right)$$

Then with the definition of $F(X)$, the assumption that $f_i(x)$ is $\beta$-smooth and Cauchy inequality, we have

$$I(\pi||\pi^{\hat{X}_1}) \leq n^2 \beta^2 \mathbf{E}_\pi \mathbf{E}_{\hat{\pi}^{\hat{X}_1=\hat{x}_1}} \|(\hat{x}_2^T, \ldots, \hat{x}_n^T)U_2^T\|^2.$$

Here $U_2 \in \mathbb{R}^{nd \times (n-1)d}$ consisting of the last $(n-1)d$ columns of $U$. It follows that

$$
\begin{aligned}
& I(\pi || \pi^{\hat{X}_1}) \\
& \leq 2n^2\beta^2 \mathbf{E}_\pi \mathbf{E}_{\hat{\pi}^{\hat{X}_1=\hat{x}_1}} \|(\hat{x}_2^T - \tilde{x}_2^T, \ldots, \hat{x}_n^T - \tilde{x}_n^T)U_2^T\|^2 \\
& \quad + 2n^2\beta^2 \mathbf{E}_\pi \mathbf{E}_{\hat{\pi}^{\hat{X}_1=\hat{x}_1}} \|(\tilde{x}_2^T, \ldots, \tilde{x}_n^T)U_2^T\|^2 \\
& \leq 2n^2\beta^2 \left( \frac{(n-1)d}{\alpha + \frac{n}{\eta}(1-\lambda_2)} + \mathbf{E}_\pi \|(\tilde{x}_2^T, \ldots, \tilde{x}_n^T)U_2^T\|^2 \right)
\end{aligned}
\tag{24}
$$

where $\{\tilde{x}_i\}$ is the minimizer of the potential of $\hat{\pi}^{\hat{X}_1=\hat{x}_1}$ that satisfies

$$
\nabla_{\hat{x}_i} F(\sqrt{n}U\hat{X})|_{\hat{X}=(\hat{x}_1, \tilde{x}_2, \ldots, \tilde{x}_n)} + \frac{n}{\eta}(1-\lambda_i)\tilde{x}_i = 0,
\tag{25}
$$

and the last line follows from Lemma 3.1 and the fact that the strong convexity constant of $\hat{\pi}^{\hat{X}_1=\hat{x}_1}$ is $\alpha + \frac{n}{\eta}(1-\lambda_2)$. Then, by equation 25, the smoothness of $f_i$, and each $f_i$ has the minimizer $m$,

$$
\begin{aligned}
& \mathbf{E}_\pi \|(\tilde{x}_2^T, \ldots, \tilde{x}_n^T)U_2^T\|^2 \\
& \leq \frac{\eta^2}{n(1-\lambda_2)^2} \mathbf{E}_\pi \|U_2 U_2^T \nabla F|_{\hat{X}=(\hat{x}_1, \tilde{x}_2, \ldots, \tilde{x}_n)}\|^2 \\
& \leq \frac{\eta^2\beta^2 n}{(1-\lambda_2)^2} \mathbf{E}_\pi \|\hat{x}_1 - m\|^2 \leq \frac{\eta^2\beta^2 d}{(1-\lambda_2)^2\alpha}
\end{aligned}
\tag{26}
$$

where $\nabla^T F = (\nabla f_1^T(x_1), \ldots, \nabla f_n^T(x_n))$ and $(x_1^T, \ldots, x_n^T)^T = \sqrt{n}U\hat{X}|_{\hat{X}=(\hat{x}_1, \tilde{x}_2, \ldots, \tilde{x}_n)}$. The first line is given by equation 25. The last line is obtained by the assumption that each $f_i$ has the minimizer $m$ and the contraction property of the proximal method. With equation 23, equation 24, equation 26 and Pinsker's inequality, we complete the proof. $\qquad\square$

Finally, we present the overall iteration complexity, assuming $\eta$ is sufficiently small. In Proposition 5.5, the iteration complexity is affected by the structure of the bipartite graph from two aspects: the number of nodes $n$ and the sparsity measured by the spectral gap $1-\lambda_2$. In Appendix A.5.2, we demonstrate that our sampler presents faster convergence than quadratic dependence on $n$ for perfect binary trees and circular graphs. In Appendix A.4, we show that the same non-asymptotic complexity can be obtained under Wasserstein distance.

**Proposition 5.5.** *For simplicity, we assume $\eta$ is small enough so that the quadratic term $\frac{\eta^2\beta^2}{(1-\lambda_2)^2\alpha}$ in Theorem 5.4 is upped bounded by the first order term $\frac{n-1}{\alpha+\frac{n}{\eta}(1-\lambda_2)}$. Under Assumption 5.1 and 5.2, to achieve $\epsilon$ TV distance to $\pi$, with an exact sampler for Algorithm 2 and $\eta = \mathbf{O}\left(\frac{\alpha\epsilon^2(1-\lambda_2)}{\beta^2 dn}\right)$, one needs iterations $K = \tilde{\mathbf{O}}\left(\frac{n^2\beta^2 d}{\alpha^2\epsilon^2(1-\lambda_2)}\right)$. Here $\lambda_2$ is the second largest eigenvalue of the doubly stochastic matrix defined in Assumption 5.2.*

*Proof.* Please see the proof in Appendix A.3, which follows the same idea in the proof of Proposition 4.4. $\qquad\square$

*Remark* 5.6 (Asymptotically convergence). We emphasize that our sampler still converges asymptotically after replacing Assumption 5.1 with mild regularity conditions. As shown in Proposition 5.5, to establish the convergence of the overall sampler, we only need to ensure the convergence of the Gibbs sampler and the convergence of $\bar{\pi}$ towards $\pi$ with a sufficiently small $\eta$. Roberts & Smith (1994) gave the simple conditions under which Gibbs samplers converge asymptotically. The second convergence can be proved by Scheffé's lemma (Scheffé, 1947), as shown in proposition 5.1 in Yuan et al. (2023) and equation 6 in Vono et al. (2022a). Both asymptotic convergences require only mild regularity conditions. Moreover, we demonstrate the convergence in Figure 2b and 2c.

# 6 Experiments

We mainly compare our sampler with two baselines developed from Langevin dynamics. D-SGLD (Gürbüzbalaban et al., 2021) is the decentralized version of stochastic gradient Langevin dynamics, which proceeds as $x_i^{k+1} = \sum_j \sigma_{ij} x_j^k + \eta \nabla \tilde{f}_i(x_i^k) + \sqrt{2\eta} w_i^{k+1}, w_i^{k+1} \sim \mathcal{N}(\mathbf{0}, \mathbf{I}_d)$ where $x_i^k$ is the $k$-th sample on node $i$. Note that here $\eta$ is fixed, while another baseline, D-SGLD with diminishing step sizes (Parayil et al., 2020), utilizes diminishing step sizes and has been proved to operate under weaker assumptions.

We compare our sampler against D-SGLD using Gaussian targets on perfect binary trees of depth 3. In this case, $\sigma_{ij} = 1/3$ if there is one edge connecting $i$ and $j$, $\sigma_{ii} = 1 - \sum_j \sigma_{ij}$, and we replace the unbiased estimation of gradients $\nabla \tilde{f}_i(x_i^k)$ by the exact value. The dimension $d$ is 5, and the initial distribution on each node is $\mathcal{N}(\mathbf{0}, \mathbf{I}_5)$. The performance is measured by the estimated 2-Wasserstein distance. We also conduct experiments on a more challenging target defined by $\exp(-\sum_{i=1}^4 f_i(x)) \propto \exp(-\|x\|^{1.5} - \|x - 0.5\|^{1.5} - \|x - 1\|^{1.5} - \|x - 1.5\|^{1.5})$ where each minimizer is an all-one vector multiplied by a scale. It is important to note that this target distribution does not satisfy any condition outlined in Assumptions 4.1 and 5.1. Each potential function has distinct minimizers and does not have properties such as strong convexity or smoothness. Due to potential bias when estimating the $W_2$ distance for this non-Gaussian target, we measure performance using the L2 distance between the estimated mean and the ground truth. We repeat each experiment three times and report the mean and 2-sigma error of measurements. Our sampler demonstrates robust performance even for distributions that do not satisfy the assumptions, and it shows greater stability in terms of initial step size compared to the two Langevin-based samplers.

We highlight the detailed observations. First, the performance of our Gibbs sampler aligns with our qualitative results in Section A.5.1: larger step size yields faster convergence but also leads to more bias. Furthermore, one advantage of our Gibbs sampler is that it converges with any step size, while D-SGLD is highly sensitive to the initial step size: a large step size can result in non-convergence (Figure 2c) or even divergence (Figure 2a). Also, D-SGLD with diminishing step sizes exhibits slow convergence when the initial step size is small or when a large number of iterations is required (Figure 2b).

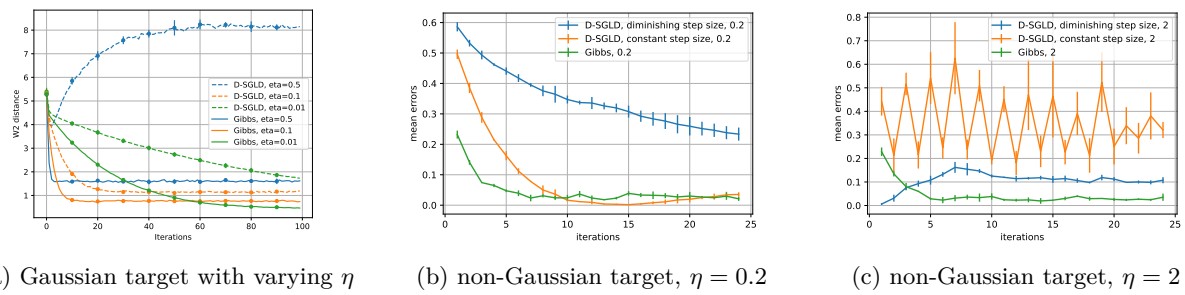

(a) Gaussian target with varying $\eta$     (b) non-Gaussian target, $\eta = 0.2$     (c) non-Gaussian target, $\eta = 2$

Figure 2: Comparison against D-SGLD with fixed step sizes and diminishing step sizes

Please see Appendix A.6 for more experiments.

# 7 Conclusions

In conclusion, our study introduces a fully distributed sampler using blocked Gibbs sampling, adaptable to various bipartite graph structures without centralized servers. The non-asymptotic analysis provides a theoretical understanding of its behavior, contributing to the theoretical exploration of fully distributed samplers. Exploring the design and analysis of a more efficient augmented distribution represents a promising direction for future research. It is also interesting to explore how to modify our sampler to mitigate practical issues, e.g., communication cost and statistical heterogeneity.

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

## A  Appendix

### A.1  Other related works

**Proximal Samplers:** Recently, the proximal sampler (Lee et al., 2021; Chen et al., 2022) achieved the state-of-the-art probable dimension dependency under different assumptions. Its idea is as follows. To sample from $\exp(-f(x))$, one can sample from $\exp(-f(x) - \frac{1}{2\eta}\|x - y\|^2)$ alternatively between $x$ and $y$ with a Gibbs sampler. Our proposed algorithm can be seen as a generalization of proximal samplers into a distributed setting. Vono et al. (2022a) presents a MCMC algorithm using ADMM-type splitting. It considers sampling from $\exp(-\sum_{i=1}^n f_i(x_i)) - \frac{1}{2\eta}\|x_i - \theta\|^2)$ with a Gibbs sampler and then control the distance between marginals and the target with a small $\eta$. Despite having similar concepts to ours, this sampler requires a centralized server for the parameter $\theta$. Combining with the proximal sampler framework, Lee et al. (2021) has proposed a high-accuracy approximate sampling algorithm for the same target $\exp(-\sum_{i=1}^n f_i(x))$. This approximate Metropolis-Hasting algorithm again requires a server node to gather information from every client.

**Distributed MCMC with server-client structures:** Embarrassingly parallel MCMC (Neiswanger et al., 2013) employs a two-step process where MCMC runs in parallel on data partitions, followed by a server combining the local results, but suffers from misrepresentation of low-density regions. To mitigate embarrassing failures, De Souza et al. (2022) proposes a new combination strategy, leveraging Gaussian Process surrogate modeling and active learning. To reach high-density regions faster, Chowdhury & Jermaine (2018) utilizes an auxiliary shepherding distribution (SD) to control parallel MCMC chains. Recently several works have focused on Bayesian federated learning. Vono et al. (2022b) extends SGLD to the federated learning setting, handling the communication bottleneck using gradient compression. Plassier et al. (2021)

introduces "DG-LMC", a novel, scalable, synchronous distributed MCMC algorithm that integrates Langevin Monte Carlo and Gibbs sampling for efficient Bayesian inference in large-scale data analysis. Kotelevskii et al. (2022) addresses uncertainty quantification of personalized federated learning with federated stochastic optimization algorithms.

**Distributed Optimization:** The typical problem in distributed optimization is to minimize the target function, $\sum_{i=1}^{n} f_i(x)$ under certain assumptions on each $f_i$ and the underlying communication graphs. A class of classical gradient descent based algorithms (Yuan et al., 2016) on undirected connected graphs can be represented as $x_{k+1}^i = \sum_{j=1}^{n} a_{ij} x_k^j - \eta_k \nabla f_i(x_k^i)$ where $x_k^i$ is the local estimated variable on client $i$, $A = (a_{ij})$ is the connection matrix, and $\eta_k$ is the step size at the $k$-th iteration. There exists a great number of distributed optimization algorithms that address unconstrained problems, e.g., the alternating direction method of multipliers (Boyd et al., 2011), the fast gradient method (Jakovetić et al., 2014), the push-sum algorithm (Nedic et al., 2017; Xi et al., 2018), and others. We recommend several surveys (Yang et al., 2019; Halsted et al., 2021; Zheng & Liu, 2022) for readers interested in more comprehensive discussions. We emphasize that a majority of distributed optimization algorithms do not rely on centralized servers.

**Stochastic Gradient Langevin Dynamics:** The centralized Stochastic Gradient Langevin Dynamics (SGLD) can be used to sample from $\exp(-\sum_{i=1}^{n} f_i(x))$ on the server-client architecture. It requires evaluating unbiased estimators of gradients by mini-batches. This method has sparked a huge number of associated works in sampling and other machine learning areas (Patterson & Teh, 2013; Li et al., 2016a; Dubey et al., 2016; Li et al., 2016b; Bardenet et al., 2017; Li & Erdogdu, 2020). Non-asymptotic bounds in terms of Wasserstein distance for strongly convex and smooth potentials have been established in Dalalyan (2017a); Dalalyan & Karagulyan (2019). It is then suggested to combine SGLD with variance reduction to reduce the variance of stochastic gradients (Zou et al., 2019; 2021). Recently, Kinoshita & Suzuki (2022); Balasubramanian et al. (2022) have established non-asymptotic bounds under weaker assumptions on potentials.

## A.2 Proof of improved convergence rates in Theorem 5.3

To begin with, we rewrite the key results from Yuan et al. (2023). The following corresponds to the proof of Theorem 10 and Proposition 16 in it. As discussed, the following lemma directly comes from utilizing the additional condition $\sum_a \sigma_{ab} \leq 1, \sum_b \sigma_{ab} \leq 1$ we add in this work.

**Lemma A.1.** *Assume Steps 3 and 4 in Algorithm 2 are exact without biased terms. Denote the distribution of the $k$-th samples by $\nu_k^{AB}$. For the target distribution $\hat{\pi}^{AB}$ equation 16, under Assumption 5.1, we have*

$$\mathrm{KL}(\nu_{k+1}^A \| \hat{\pi}^A) \leq \exp\left(-\eta \int_0^1 c_t \mathrm{d}t\right) \mathrm{KL}(\nu_k^B \| \hat{\pi}^B) \tag{27}$$

*where*

$$c_t = \left[\frac{\eta t(1-t)}{\min_b \sum_a \sigma_{ab}} + (1-t)^2 \frac{h}{\alpha} + \frac{t^2}{\alpha}\right]^{-1}.$$

*Here $h$ is the upper bound of eigenvalues of $H^T H$ where the entry of $H$ at the $a$-th row and $b$-th column is $\sigma_{ab}/\sum_a \sigma_{ab}$. In the proof of Proposition 16 in Yuan et al. (2023), this $H$ matrix is denoted as $A$. We rewrite it to avoid conflicts between notations.*

Based on Lemma A.1, we have the proof of Theorem 5.3.

*Proof.* We consider the estimation of $h$ in Lemma A.1. As a comparison, Yuan et al. (2023) bounds $h$ by the squared operator norm of $H$, which follows that

$$\|H\|_{\mathrm{op}}^2 \leq |A||B| \|H\|_{\max}^2 \leq |A||B|.$$

In contrast, we notice

$$\|H\|_{\mathrm{op}}^2 \leq \|H\|_F^2 = \sum_a \sum_b \frac{\sigma_{ab}^2}{(\sum_a \sigma_{ab})^2} \leq \sum_a \sum_b \frac{\sigma_{ab}}{(\sum_a \sigma_{ab})} \leq |B|.$$

$|A|$ and $|B|$ are the number of nodes in the group $A$ and $B$, respectively.

This modification leads to the new lower bound for $c_t$ and hence the improved convergence rate shown in Theorem 5.3. The contraction from $\text{KL}(\nu_{k+1}^A || \hat{\pi}^A)$ to $\text{KL}(\nu_{k+1}^B || \hat{\pi}^B)$ can be analyzed in the same way. $\qquad\square$

### A.3 Proof of Proposition 5.5

*Proof.* The proof is generalized from the proof of Proposition 4.4. With Theorem 5.3, the required iteration to ensure the convergence of Algorithm 2 is $K = \tilde{O}\left(\frac{n}{\eta\alpha}\right)$. Apply our assumption that the quadratic term is bounded by the first order term to Theorem 5.4, then we have $\text{TV}(\bar{\pi}, \pi) = O\left(\sqrt{\frac{n\beta^2 d\eta}{\alpha(1-\lambda_2)}}\right)$. This implies that to have $\text{TV}(\bar{\pi}, \pi) \leq \epsilon/2$, $\eta = O\left(\frac{\alpha\epsilon^2(1-\lambda_2)}{\beta^2 dn}\right)$. Combining all the requirements, we conclude the proof. $\qquad\square$

### A.4 Iteration complexity of Algorithm 2 under Wasserstein distance

In proposition 5.5, we established the interaction complexity when the distance is measured by TV distance. The same complexity can be obtained under Wasserstein distance. Our complexity measured by TV distance is based on Theorem 5.3 and Theorem 5.4. Both theorems can be reformulated with KL divergence: on Theorem 5.4, this can be done by omitting Pinsker's inequality used in the last step. Since we assume each potential is strongly convex, we can apply Talagrand inequality (Otto & Villani, 2000) on $\pi$ and $\hat{\pi}$. This inequality claims that the Wasserstein distance is upper bounded by the KL divergence. Hence, the same iteration complexity is followed by the triangle inequality of Wasserstein distance.

### A.5 Experiments

#### A.5.1 Qualitative analysis on a general Tree

To illustrate how our algorithm works for general trees, we conduct experiments on a tree with five nodes shown in Figure 3. Each $f_i$ represents a 5-dimensional Gaussian distribution with a randomly generated mean and covariance matrix. We denote each one as $N(\mu_i, \Sigma_i)$, and our target is $\exp(-\sum_{i=1}^5 f_i(x)) \propto N(\tilde{\mu}, \tilde{\Sigma})$ where $\tilde{\mu} = \tilde{\Sigma}^{-1}(\sum_{i=1}^5 \Sigma_i^{-1}\mu_i)$ and $\tilde{\Sigma}^{-1} = \sum_{i=1}^5 \Sigma_i^{-1}$. All the Gaussian distributions have condition numbers (the ratio of smoothness and strong convexity) ranging from 2 to 5. The initial samples of $X_1$, $X_4$, and $X_5$ are drawn from a standard Gaussian $N(0, \mathbf{I})$ independently. As explained in Algorithm 2, at each step, our sampler enables fully distributed sampling. For instance, clients 2 and 3 can sample from conditionals independently once $x_1$, $x_4$, and $x_5$ are given. Each sampler runs for 400,000 steps, with the first 100,000 steps being the burn-in stage. In Figure 4, we compare the histogram of generated samples and the true density along the first coordinate. In this experiment, we can demonstrate the following points.

- Our sampler can operate on general trees, accommodating scenarios where each node may possess unique strong convexity and smoothness parameters. We point out that a reasonable choice in this case is $\eta = 0.001$ and $K = 400,000$ (See the second column in Figure 4).

- All marginals converge to the same distribution. This observation holds even when all marginals converge to a biased distribution as shown in the first column of Figure 4.

- In practice, we recommend users identify the source of errors and then tune the step size $\eta$ or iterations $K$ accordingly: reducing $\eta$ may require more iterations. More specifically, if the value of $K$ is small, alternative diagnostic tools, such as trace plots, can be employed to assess the convergence of the blocked Gibbs sampler. This corresponds to the third column of Figure 4. If $\eta$ is relatively big, it becomes apparent that all marginals converge to a common distribution, albeit one that is biased toward our target.

#### A.5.2 Quantitative analysis

In this section, we assume each $\exp(-f_i(x))$ is $\mathcal{N}(\mathbf{1}, \mathbf{I}_d)$, and the initial distribution for each node is $\exp(-f_i(x))$ is $\mathcal{N}(\mathbf{0}, \mathbf{I}_d)$. Also, following the Metropolis weights, we set $\sigma_{ab} = 1/(\max(h_a, h_b)+1)$ if there is an edge between

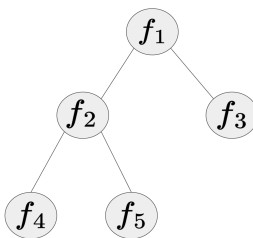

Figure 3: Each local distribution $\exp(-f_i(x))$ is a 5-dimensional Gaussian

node $a$ and $b$. Here, $h_a$ is the degree (the number of neighbors) of node $a$. For these two network structures considered below, perfect binary trees (with layers more than 2) and circular graphs (each node connects to its two neighbors), the non-zeros value of $\sigma_{ab}$ is $1/3$. In this section, we repeat each of our experiments under five different random seeds and show the 2-sigma error bar in all figures and tables. It supports that our findings are statistically significant.

**Dependency on the number of nodes $n$**

In Proposition 5.5, we have proved the dependency of the mixing time $K$ on the number of nodes $n$. Note that it is unclear if this upper bound is tight. To better understand the complexity of $n$, we conduct experiments on two graph structures, perfect binary trees and circular graphs. We run 100 independent chains for 40 iterations and then estimate the empirical mean and covariance matrix at each iteration. Since the target distribution is Gaussian, we compute the $W_2$ distance with the explicit formula of $W_2$ between two Gaussian distributions.

In Figure 5, we run our sampler with layers ranging from 3 to 6 corresponding to $n = 7, 15, 31$, and 63, respectively. Then, we repeat our experiments on circular graphs with the same $n$. These figures show our sampler converges even when the graph is sparse and the number of nodes is relatively large. We further analyze the curves from two perspectives: the slope after the logarithm transformation and the minimal distance along a chain. By Theorem 5.3, the estimated slope after applying logarithm transformation is the convergence rate of the Gibbs sampler. In Table 1, we observe that the convergence rate is small for large $n$. Moreover, it may indicate a faster convergence rate than linear rates on $n$. We also estimate the biased term discussed in Theorem 5.4 by the minimal $W_2$ over each chain in Table 2. Again, we observe that the biased term is large for graphs with more nodes as indicated in Theorem 5.4. Furthermore, the coefficient of determination ($R^2$) for fitting the distances and $\sqrt{n}$ is approximately 0.940; this may indicate that the dependency in this setting aligns with our theoretical analysis in Theorem 5.4.

Table 1: Estimated slopes represented by the mean and 2-sigma error

| Graphs | $n = 7$ | $n = 15$ | $n = 31$ | $n = 63$ |
|---|---|---|---|---|
| Perfect binary trees | $-0.054 \pm 0.003$ | $-0.047 \pm 0.002$ | $-0.039 \pm 0.002$ | $-0.034 \pm 0.001$ |
| Circular graphs | $-0.056 \pm 0.003$ | $-0.038 \pm 0.002$ | $-0.028 \pm 0.002$ | $-0.024 \pm 0.001$ |

Table 2: Minimal distances represented by the mean and 2-sigma error

| Graphs | $n = 7$ | $n = 15$ | $n = 31$ | $n = 63$ |
|---|---|---|---|---|
| Perfect binary trees | $0.170 \pm 0.043$ | $0.270 \pm 0.020$ | $0.376 \pm 0.024$ | $0.457 \pm 0.035$ |
| Circular graphs | $0.187 \pm 0.013$ | $0.367 \pm 0.016$ | $0.526 \pm 0.015$ | $0.637 \pm 0.025$ |

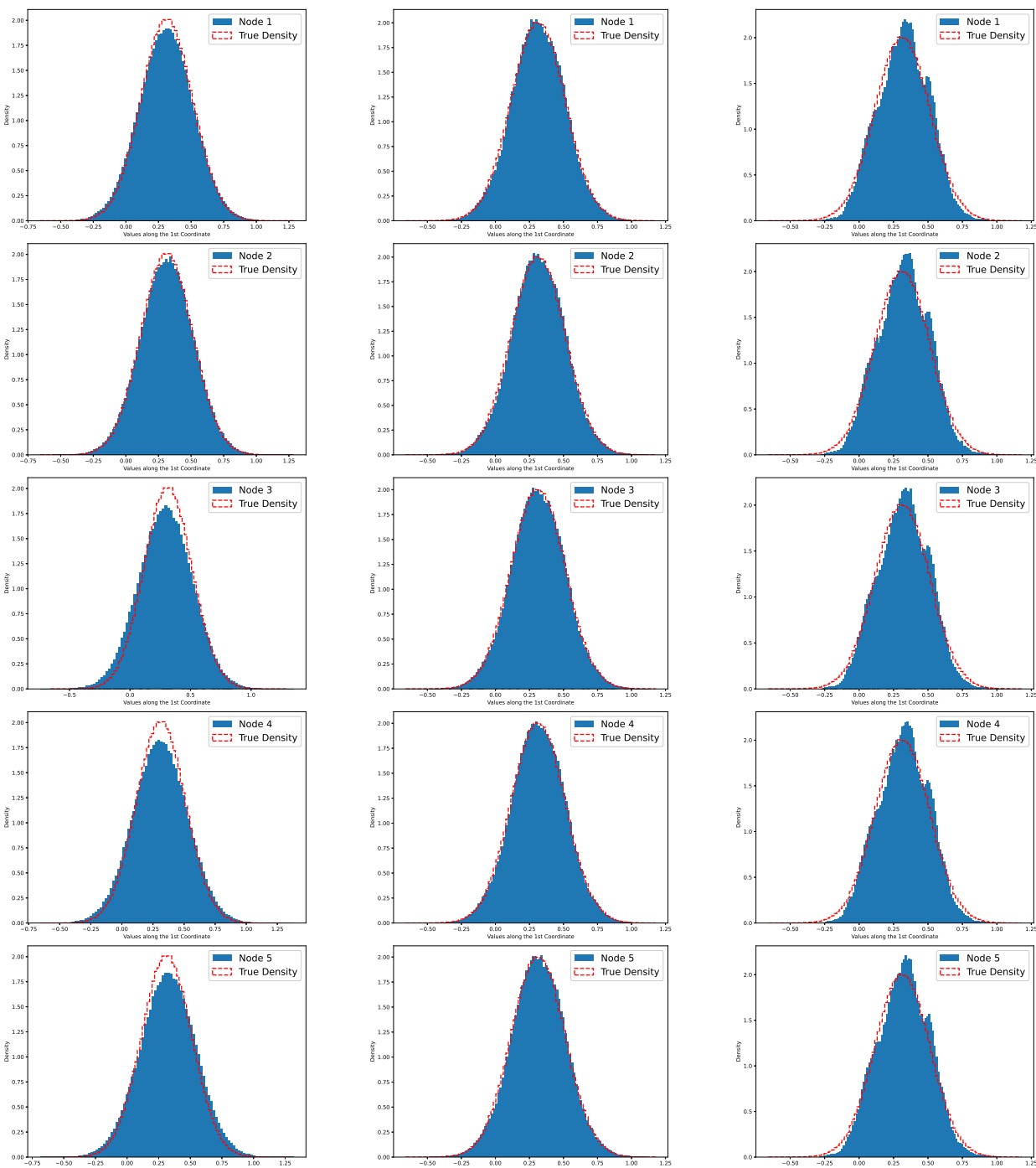

Figure 4: Each column in the figure represents the histogram of values on all five nodes with unique $\eta$ values. The columns are arranged from left to right, where the $\eta$ values are 0.01, 0.001, and 0.0001, respectively. In the first column, $\eta$ is too large that all marginals converge to a biased distribution, while in the third column, $\eta$ is too small that users need more iterations.

## A.6 Comparison to D-ADMMS (Tzikas et al., 2024)

D-ADMM is a distributed MCMC variant of ADMM (Mateos et al., 2010) that treats the primal variable of each agent as the sample and, at each iteration, performs a noisy proximal update with neighbor averaging.

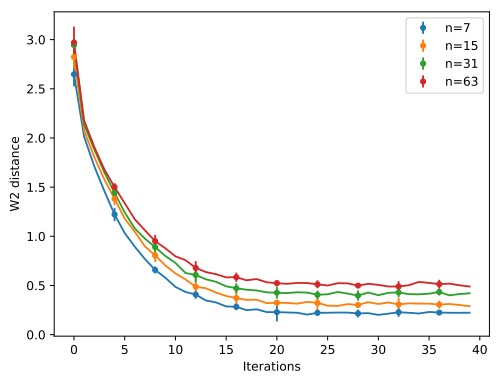

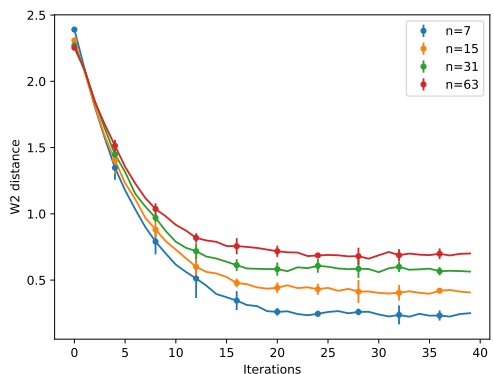

(a) Convergence on perfect binary trees

(b) Convergence on circular graphs

Figure 5: The mean and 2-sigma error bar of estimated $W_2$ distances under various nodes on trees (Left) and circular graphs (Right)

Table 3: Gaussian target on a three-layer perfect binary tree.

| Iterations | 0 | 20 | 40 | 60 | 80 |
|---|---|---|---|---|---|
| D-ADMMS with $\rho$=0.01 | 5.83 ± 0.06 | 13.37 ± 0.30 | 13.52 ± 0.01 | 13.57 ± 0.09 | 13.38 ± 0.06 |
| D-ADMMS with $\rho$=0.1 | 5.81 ± 0.09 | 9.75 ± 0.14 | 9.65 ± 0.13 | 9.57 ± 0.12 | 9.53 ± 0.01 |
| D-ADMMS with $\rho$=0.5 | 5.83 ± 0.02 | 4.38 ± 0.03 | 4.39 ± 0.02 | 4.43 ± 0.08 | 4.39 ± 0.08 |
| D-ADMMS with $\rho$=5 | 5.84 ± 0.03 | 2.76 ± 0.01 | 1.51 ± 0.02 | 0.97 ± 0.01 | 0.76 ± 0.01 |
| D-SGLD with $\eta$=0.5 | 5.84 ± 0.03 | 8.64 ± 0.02 | 9.96 ± 0.07 | 10.26 ± 0.04 | 10.31 ± 0.13 |
| D-SGLD with $\eta$=0.1 | 5.86 ± 0.02 | 1.67 ± 0.03 | 1.50 ± 0.01 | 1.48 ± 0.06 | 1.53 ± 0.01 |
| D-SGLA with $\eta$=0.01 | 5.83 ± 0.06 | 4.71 ± 0.02 | 3.86 ± 0.02 | 3.17 ± 0.03 | 2.63 ± 0.03 |
| **Ours** with $\eta$=0.5 | 5.83 ± 0.06 | 2.11 ± 0.02 | 2.05 ± 0.01 | 2.10 ± 0.01 | 2.09 ± 0.06 |
| **Ours** with $\eta$=0.01 | 5.83 ± 0.06 | 3.03 ± 0.03 | 1.63 ± 0.02 | 0.98 ± 0.04 | 0.76 ± 0.01 |

**Gaussian study** To stress test robustness, we repeated the Gaussian experiment on the same three-layer perfect binary tree as in Figure 2a but increased the dimensionality from $d$=5 to $d$=32 and replaced the Gaussian initialization with a uniform initialization. The penalty parameter $\rho$ in D-ADMMS was varied. We report mean ± std over three trials in Table 3.

The results in Table 3 show a pronounced hyperparameter sensitivity for D-ADMMS. D-SGLD performs worse overall: at $\eta = 0.5$ it even degrades over time and while $\eta = 0.1$ is more stable, it still plateaus higher . In contrast, our sampler is robust across step sizes: both configurations converge without instability, with $\eta = 0.01$ achieving 0.76 at 80 iterations matching the best D-ADMMS setting and $\eta = 0.5$ remaining stable around 2.09. These trends indicate that our method attains competitive accuracy with substantially less hyperparameter sensitivity.

**Non-Gaussian study** We also evaluated D-ADMMS on the non-Gaussian target of Figure 2b and compared against our sampler. The performance metric is the $L_2$ distance between the estimated mean and the ground-truth mean. Results appear below in Table 4.

Table 4 shows that D-ADMMS again exhibits strong hyperparameter sensitivity: with a small penalty ($\rho$=0.2) the $L_2$ mean error oscillates and does not settle, whereas larger penalties improve monotonicity but still remain high at 25 iterations. In contrast, our sampler is robust across step sizes: both $\eta$=0.2 and $\eta$=2 reach low error quickly and stay low thereafter, with the best value 0.01 at 25 iterations. Overall, our method achieves lower error with markedly less tuning effort on this non-Gaussian target.

Table 4: Non-Gaussian target

| Iterations | 5 | 10 | 15 | 20 | 25 |
|---|---|---|---|---|---|
| D-ADMMS with $\rho$=0.2 | $0.03 \pm 0.01$ | $0.07 \pm 0.05$ | $0.03 \pm 0.025$ | $0.08 \pm 0.06$ | $0.06 \pm 0.02$ |
| D-ADMMS with $\rho$=2 | $0.42 \pm 0.02$ | $0.28 \pm 0.03$ | $0.20 \pm 0.01$ | $0.15 \pm 0.01$ | $0.11 \pm 0.01$ |
| D-ADMMS with $\rho$=5 | $0.55 \pm 0.003$ | $0.46 \pm 0.007$ | $0.39 \pm 0.013$ | $0.32 \pm 0.005$ | $0.27 \pm 0.005$ |
| **Ours** with $\eta$=0.2 | $0.05 \pm 0.006$ | $0.02 \pm 0.011$ | $0.02 \pm 0.003$ | $0.02 \pm 0.009$ | $0.01 \pm 0.002$ |
| **Ours** with $\eta$=2 | $0.05 \pm 0.003$ | $0.03 \pm 0.014$ | $0.02 \pm 0.008$ | $0.02 \pm 0.003$ | $0.03 \pm 0.012$ |

### A.7 Revisiting the differences to Yuan et al. (2023)

Both this work and Yuan et al. (2023) employ Gibbs updates, but the problem setting and—crucially—the guarantees we establish are different.

**Scope of the target and augmented model.** We focus on the fully distributed composite target $\pi(x) \propto \exp\big( - \sum_{i=1}^n f_i(x)\big)$ and analyze a *client-only* blocked Gibbs sampler over bipartite graphs for the augmented joint law

$$\hat{\pi}(x_1, \ldots, x_n) \; \propto \; \exp\left(-\sum_{i=1}^n f_i(x_i) \; - \; \sum_{a \in A} \sum_{b \in B} \frac{\sigma_{ab}}{2\eta} \|x_a - x_b\|^2 \right).$$

In Yuan et al. (2023), the *augmented* distribution is itself the target of inference: their results establish mixing of a Gibbs chain for that augmented law, but do not relate it non-asymptotically to the composite objective $\exp(-\sum_i f_i(x))$. In contrast, our augmented law is used only as an *instrument*: we prove that for sufficiently small $\eta$ the induced marginal distribution is close to the composite target $\pi$—see Theorems 4.3 and 5.4—thereby bridging augmented and original targets with explicit TV bounds.

**New non-asymptotic bias control** A central novelty of this paper is *quantifying* the bias induced by augmentation:

- **Two-node composite:** Theorem 4.3 gives an explicit $\mathrm{TV}(\pi, \pi^X)$ bound—absent in Yuan et al. (2023).

- **Networks:** Theorem 5.4 bounds $\mathrm{TV}(\bar{\pi}, \pi)$ for the average $\bar{\pi}$ induced by $\hat{\pi}$, with explicit dependence on the spectral gap $1 - \lambda_2$ of the communication weights. This bridges the augmented sampler to the *original* composite target on graphs.

These results enable end-to-end iteration complexities (Propositions 4.4 and 5.5) in total variation, which were not available in Yuan et al. (2023).

**Sharper Gibbs mixing for the augmented chain.** We revisit the block-Gibbs analysis of Yuan et al. (2023) and obtain a tighter contraction constant by (i) imposing the natural weight constraints $\sum_a \sigma_{ab} \leq 1$ and $\sum_b \sigma_{ab} \leq 1$, and (ii) tightening the bound on the intermediate matrix term. The resulting Theorem 5.3 improves the convergence rate constant by up to $\mathcal{O}(n^2)$ compared to Theorem 10 of Yuan et al. (2023), yielding faster mixing of the augmented chain and directly strengthening the overall complexity bounds.

