# OpenReview forum: "Client-only Distributed Markov Chain Monte Carlo Sampling over a Network"
_TMLR — Accepted by TMLR_

### Review · Reviewer_GcEm · 2025-04-15

**Summary Of Contributions:**

This paper studies the problem of sampling from a target distribution $\pi(x) \propto \text{exp}(-\sum_{i=1}^n f_i(x_i \mid \mathcal{D}_i))$, where each function $f_i$ depends on local data $\mathcal{D}_i$ by client $i$. The key contribution in this work is to provide non-asymptotic analysis to an existing distributed MCMC algorithm (Yuan et al. 2023) without a central server, relying only on client communication in a two-node case and extension to bipartite graph. In particular, the authors claim that

1. They propose a new distributed sampling algorithm that is fully operational on any bipartite graphs.
2. They improve the existing non-asymptotic bound from Yuan et al. (2023) and provide the non-trivial upper bound between marginals $\hat \pi$ and $\pi$.
3. They conduct empirical comparison against D-SGLD and illustrate the enhanced stability and performance of the proposed algorithm under specific conditions.

**Audience:**

Yes

**Broader Impact Concerns:**

No specific negative ethical implications or broader impact concerns requiring a dedicated statement were identified from the provided materials. The research is primarily algorithmic.

**Claims And Evidence:**

No

**Requested Changes:**

1. The authors should provide a detailed comparison of their results with those presented by Yuan et al. (2023). It is crucial that they clearly indicate the necessity and significance of their findings, as Yuan et al. (2023) have already performed non-asymptotic analyses for both Algorithms 1 and 2, and established their rate of convergence. Without such clarity, the importance of the authors' theoretical contributions remains unclear.
2. More discussion and comparison of earlier distributed MCMC algorithms is needed. Currently, the motivation for selecting Gibbs sampling is not adequately addressed, particularly when more advanced alternatives, such as SGLD or HMC, have been overlooked.
3. The empirical evaluation needs to be strengthened, including state-of-the-art baselines beyond D-SGLD. The authors should test on more high-dimensional problems and analyze performance across different network topologies beyond bipartite graphs to numerically see the robustness of their algorithms.

**Strengths And Weaknesses:**

### Strengths
1. The inherent client-only communication pattern enhances privacy and robustness compared to server-dependent architectures.
2. The theoretical analyses are rigorous.

### Weaknesses
1. The theoretical findings are incremental compared with Yuan et al. (2023). Although the authors claim they use the novel techniques in their non-asymptotic bound, I do not find where the authors discuss their novelty, even after reading the proofs of Theorem 4.3 and Theorem 5.4.
2. The novelty and practical significance were questioned relative to existing distributed MCMC and optimization literature, where the discussion about the previous works is less clear.
3. Experiments lack comparisons with a wider set of relevant baselines and do not explore sufficiently diverse scenarios (models, data, network structures beyond bipartite graphs) to fully demonstrate the claimed advantages.

---

### Review · Reviewer_r5tu · 2025-04-16

**Summary Of Contributions:**

This manuscript introduces a fully distributed Markov Chain Monte Carlo (MCMC) sampler for Bayesian inference in decentralized networks, which eliminates the need for centralized servers. The target distribution is defined as $\pi \propto \exp\left(-\sum_{i=1}^n f_i(x)\right)$, where each $f_i$ corresponds to local data at client $i$. The key innovation is a block Gibbs sampling approach on an augmented distribution that incorporates pairwise quadratic terms between clients on a bipartite graph, which enables decentralized communication. Non-asymptotic convergence guarantees in Total Variation (TV) distance for the distributed Gibbs sampler to address a gap in the analysis of fully decentralized samplers.

**Audience:**

Yes

**Broader Impact Concerns:**

The work promotes privacy-preserving Bayesian inference in distributed systems, which is crucial for applications like healthcare, finance, and edge computing, where data cannot be centralized. The lack of a centralized server reduces single points of failure and enhances robustness, aligning with trends toward decentralized technologies.

The paper does not explicitly discuss potential downsides. For instance, in sparse graphs, the increased communication complexity could pose a limitation. The energy costs for edge devices also warrant consideration. A brief Broader Impact Statement acknowledging the trade-offs between decentralization and resource usage is advisable. Limitations in real-world deployment, such as graph connectivity requirements, should also be discussed to provide a balanced perspective.

**Claims And Evidence:**

Yes

**Requested Changes:**

1. Add a comparative discussion of prior works, explaining their limitations (e.g., server dependence, convergence guarantees) and how the proposed method addresses these. Clarify the difference between "fully distributed" (no server) and "decentralized" frameworks used in other studies.
2. For experiments, include runtime comparisons between the Gibbs sampler and baselines (e.g., D-SGLD) to address potential efficiency concerns. Discuss initialization strategies in experiments, testing how random vs. informed initializations affect convergence (as in Figure 2).
3. In Equation (5) (page 5), the expectation $\underset{x \sim \pi}{\mathbf{E}}$ is not clearly linked to the variable $x$ in $\nabla \log \frac{\pi}{\pi^X}$.

**Strengths And Weaknesses:**

**Strengths**:
1. The use of block Gibbs sampling on an augmented distribution enables true decentralization, a critical advance for privacy-sensitive applications (e.g., edge computing, sensor networks).
2. The non-asymptotic analysis is overall sound to me. It builds on strong convexity/smoothness assumptions and LSI to derive convergence rates.
3. Experiments highlight the sampler’s robustness to large step sizes (unlike D-SGLD, which diverges) and its scalability to complex graph structures.

**Weaknesses**:
1. The introduction does not sufficiently explain why Gibbs sampling is uniquely suited for distributed settings compared to other MCMC methods (e.g., Metropolis-Hastings, Langevin Monte Carlo). The problem’s relevance to Bayesian learning and privacy-preserving inference is mentioned but not deeply motivated.
2. Section 2 lists numerous works but lacks critical analysis. There is no clear mapping of how the proposed method addresses the limitations of prior distributed MCMC.
3. Figure 2 compares metrics (Wasserstein distance, mean error) under fixed iterations but not runtime. Since Gibbs sampling may have higher per-iteration costs than SGLD, a comparison of performance vs. wall-clock time is missing.
4. The paper does not discuss how initial samples are generated (e.g., random initialization, prior knowledge) or their impact on convergence. The initial distribution $\nu_0^X$ is mentioned in Proposition 4.4 but not validated in experiments.

---

### Review · Reviewer_rgxK · 2025-05-13

**Summary Of Contributions:**

The authors are tackling the problem of sampling from a probability distribution having a composite potential, where each potential function comes from a likelihood defined locally on a given node of a network of clients.
To address this issue, they propose to rely on an augmented probability distribution which is then targeted using a block Gibbs sampler. Non-asymptotic bounds with the total variation distance are derived, mostly building on the previous work by Yuan et al. (2023).
Some experiments are provided to illustrate the benefits of the proposed approach.

**Audience:**

Yes

**Claims And Evidence:**

No

**Requested Changes:**

Please see my major comments.

**Strengths And Weaknesses:**

Please find below my major comments:

**Major comments**
1) The augmented distribution $\pi^{XY}$ and the discrepancy between its marginals and the initial distribution $\pi$ has also been studied in previous works, see e.g.
* https://jmlr.org/papers/v23/20-357.html
* https://proceedings.mlr.press/v139/plassier21a.html

Could the authors compare their bounds with these works? It is not clear to me at which point results in Theorem 4.3 and Proposition 4.4 are novel.

2) Overall, there is a lack of rigor in the theoretical statements and proofs. As an example, statement and proof of Proposition 4.4 needs more rigor: you cannot just say that the second order term is supposed to vanished compared to the first-order team. I would like to see more quantitative results involving exact bounds on $\eta$ and only at the end, considering asymptotic arguments to provide order of magnitudes. Same remark for Theorem 5.3, $k$ is not defined in the statement.

3) The assumption requiring the potential functions to share the same minimizer $m$ is difficult to meet in a practical scenario as the local datasets of each client are probably different and hence the local minimizer will change. Even if the authors are stating that convergence holds without this assumption, it would be better for the paper quality to get rid of this assumption directly in the main paper.

4) With the current paper writing, it is not clear to me what is the incremental contribution compared to Yuan et al. (2023), given that all the non-asymptotic results initially came from the latter paper. The authors are saying that the bounds are tighter but do not give any comparison or discussion regarding this.

**Minor comments**
1) Introduction $\pi$ should be written $\pi(x)$ to be consistent with the right-hand side. Same remark applies throughout the whole paper, see e.g. Equation (1) and Assumption 4.1.
2) Notations are not consistent. For instance in Assumption 4.1., the common minimiser is denoted $m$ while $x^*$ is used before.

---

### Decision · Action_Editor_Wsty · 2025-07-16

**Recommendation:** Accept with minor revision

**Additional Comments:**

There has been some debate regarding the acceptance of this paper. Specifically, some reviewers felt that the contribution, relative to Yuan (2023), is rather modest. While I do not fully agree with this assessment, I found it necessary to consult Yuan (2023) myself in order to clearly understand the added value of the present work. As currently presented, I agree with the other reviewers that the distinction between this paper and Yuan (2023) is not sufficiently clear.

Therefore, I request that the authors, as part of a minor revision, include a detailed comparison between their setting and that of Yuan (2023) in the appendix. Also, referring to the entire procedure as a "new algorithm" is potentially misleading, since only Step 1 appears to be novel. I recommend that the authors group Steps 2 through 4 into a subroutine and explicitly acknowledge that this subroutine follows the method introduced in Yuan (2023).

**Audience:**

Yes

**Audience Explanation:**

Distributed MCMC computation is a timely and compelling topic. Building on the work of Yuan (2023), this paper proposes a novel scheme to address this problem, supported by theoretical guarantees. For these reasons, I believe it will be of interest to the TMLR audience.

**Claims And Evidence:**

Yes

**Claims Explanation:**

This paper presents a distributed MCMC sampler for Bayesian inference in decentralized networks. The target distribution is defined as $\pi(x) \propto \prod_{i=1}^N \pi_i(x)$, where each local factor $\pi_i$ corresponds to data held by client $i$. The core idea is to approximate this distribution using an augmented model that introduces pairwise quadratic interactions between clients connected through a bipartite communication graph. Building on the work of Yuan (2023), the authors considers a block Gibbs sampler that efficiently samples from this proxy distribution in a decentralized manner. They provide non-asymptotic convergence guarantees in total variation distance, and preliminary numerical experiments support the theoretical results.